# The Effect of Sadness on Visual Artistic Creativity in Non-Artists

**DOI:** 10.3390/brainsci13010149

**Published:** 2023-01-15

**Authors:** Massimiliano Palmiero, Laura Piccardi, Marco Giancola, Raffaella Nori, Paola Guariglia

**Affiliations:** 1Department of Communication Sciences, University of Teramo, 64100 Teramo, Italy; 2Department of Psychology, Sapienza University of Rome, 00185 Rome, Italy; 3IRCCS Ospedale San Raffaele, 00163 Rome, Italy; 4Department of Biotechnological and Applied Clinical Sciences, University of L’Aquila, 67100 L’Aquila, Italy; 5Department of Psychology, University of Bologna, 40127 Bologna, Italy; 6Department of Human Science and Society, Kore University of Enna, 94100 Enna, Italy

**Keywords:** mood induction, drawing, creativity, divergent thinking, emotion, sadness, valence, music

## Abstract

The study of the relationships between mood and creativity is long-standing. In this study, the effects of mood states on artistic creativity were investigated in ninety non-artist participants. Mood states were induced by instructing participants to listen to self-selected happy, sad, or neutral music for ten minutes. Then, all participants were asked to make two artistic drawings. To check for mood manipulation, the Profile of Mood States (POMS) was administered before and after listening to the self-selected music. After the mood induction, the negative group reported higher scores than the other two groups in the ‘depression’ subscale and lower scores than the other two groups in the ‘vigour’ subscale of the POMS; the positive mood group showed more vigour than the negative mood group. Yet, three independent judges assigned higher ratings of creativity and emotionality to the drawings produced by participants in the negative mood group than drawings produced by participants in the other two groups. These results confirmed that specific negative mood states (e.g., sadness) positively affect artistic creativity, probably because participants are more likely to engage in mood-repairing. Limitations and future research directions are presented.

## 1. Introduction

Mood is one of the most important predictors of creativity. It can either directly affect creativity or serve as an intermediary state between situational/personality factors and creative performance [1]. Mood involves low-intensity subjective, enduring and pervasive feeling states that are not caused by a specific stimulus, and can be manipulated [2,3]. Mood states are determined by valence, arousal and regulatory focus. Valence refers to the pleasantness (positive) or unpleasantness (negative) dimension, whereas arousal entails the intensity of the mood states, which can be high (activating) or low (deactivating), regardless of the valence [4]. The regulatory focus [5] refers to two motivational systems that underlie approach–avoidance behaviour, involving the way individuals balance their pursuit of pleasure and achievement with their avoidance of loss, pain, or displeasure. Specifically, it can be distinguished in terms of: promotion focus, which emphasises the pursuit of gains, such as pleasure and accomplishments, and is associated with ambition and hope; prevention focus, which emphasises the avoidance of loss or pain, and is associated with safety, security and responsibility. Both can be defined in terms of successful or unsuccessful attainment.

Overall, positive activating mood states associated with a promotion focus (e.g., happiness) produce more creativity (fluency and originality) than positive deactivating mood states associated with a prevention focus (e.g., calmness) [1]. Additionally, negative deactivating mood states are unrelated to creativity when associated with a promotion focus (e.g., sadness), whereas negative activating mood states, associated with a prevention focus (e.g., fearlessness), decrease creativity when assessed in terms of flexibility [1]. Yet, fluency and originality benefit from positive activating mood states because of enhanced flexibility (the ability to find ideas across many categories), as well as from negative activating mood states because of enhanced persistence (the ability to find ideas within a low number of categories), as compared with deactivating mood states [6]. These results led to the dual pathway model [6], which posits that dispositional (mood states) or situational variables may influence creativity through their effects on flexibility, persistence, or both. Specifically, arousal would be the precondition for creativity, whereas valence would determine the pathway by which creativity is achieved.

Other accounts focused mostly on positive or negative mood states. On the one hand, the mood-congruent retrieval theory [7] posits that positive mood states enhance the accessibility and recall of positive material in memory [8], improving cognitive organization and finding more associations between remotely connected elements. Similarly, the cognitive tuning model [9] assumes that mood states signal information content: positive mood states lead to experiencing the task as free of problems and satisfactory, whereas negative mood states lead to experiencing the task as problematic and motivate action in order to remove the threat of negative outcomes. This model also assumes that positive mood states lead people to engage in a heuristic style of information processing, whereas negative mood states lead to an analytic style of information processing. For the hedonic contingency theory [10], happy people are interested in maintaining their positive mood, which enhances individual engagement in creative activities. On the other hand, some studies revealed that negative mood states are associated with enhanced creative problem-solving, involving convergent thinking [11]. In addition, negative mood is related to the later phases of divergent thinking, that is, to the ideas that come to mind some minutes after the start of the idea-generation process [12]. Then, when positive mood is already high, for example, in supportive contexts, negative mood is strongly associated with employees’ creativity as assessed by supervisors’ ratings [13]. Alternatively, the hedonic contingency theory [10] stresses that sad people are only interested in overcoming their negative mood. This means that sadness increases the fluency of divergent thinking, resulting from motivated strategies (or mood repair) when the task is instrumentally interesting [14,15]. Then, the mood as-input model [16] assumes that the information content conveyed by mood states interacts with goal attainment. Whereas positive mood states inform that enough effort has been made, leading to a favourable evaluation of the performance, and to the possibility that people stop working, negative mood states inform that the task is not yet completed, leading to persisting in working until the optimum level of creativity is achieved. Thus, depressed artists or people in a negative mood would persist in their efforts in order to achieve high standards of creativity rather than to repair their mood. Accordingly, negative mood states positively affect creative performance when individuals perceive that recognition, rewards for creative performance, and the clarity of their feelings are high [17].

In summary, while positive mood states affect creativity in a relatively stable fashion, the effects of negative mood states on creativity appear more contextdependent, with the possibility that some tasks might not be appropriate to capture the effects of negative mood states on certain facets of creativity [18].

The majority of studies explored mood–creativity relationships using divergent thinking tasks, which involve a process perspective [19]. Although such tasks also capture the quality of responses (e.g., originality), they are based on ideational fluency [20]. They require the generation of alternative ideas to solve open-ended or ill-defined problems [21], regardless of the ability to shift between categories (flexibility) or to deeply examine a single category (persistence) [22]. Thus, using divergent thinking tasks, mood states capture mostly the quantitative aspects of divergent thinking [23].

Nevertheless, creativity can also be considered within the product perspective (e.g., paintings, collages), with the implication that quality aspects (e.g., creativity, aesthetics) are not contaminated by productivity [24,25] and that negative mood states are more effective. In this vein, previous research highlighted that the creative achievements and behaviours of eminent creative figures are affected by mood disorders; specifically, mood bipolar disorder increases creativity in the verbal and performance (acting and musical) domains [26]. Even the artistic creativity of healthy people with no experience in art improves when high biological vulnerability (levels of an adrenal steroid, which is linked to depression) is associated with a strong social rejection situation [27]. The effects of positive mood states on artistic creativity are sparser. For example, children produce more creative, energetic and technically proficient drawings after hearing familiar children’s songs than after hearing sad music [28].

Based on this evidence, the current study explored the extent to which positive and negative mood states induced by music can affect visual artistic creativity. Although there are different music mood induction procedures [29,30,31,32], in this study, self-selected music by participants was used [33,34]. This procedure allows listeners to explore emotions in familiar ways [33], inducing mood states through contagion and episodic memory [34].

Therefore, in this study, three conditions were used in order to induce sad, happy and neutral mood states. Specifically, participants in each condition selected music they believed would induce the emotion that was specified for them. Afterwards, participants were instructed to make two artistic drawings, which were evaluated in terms of creativity, emotionality and aesthetic value. This allows for analysing creative production also through an emotional or a hedonic tone of object perception compared to the only judgment of creativity [35].

Since the majority of evidence showed that artistic creativity is associated with affective disorders or negative mood states, the main hypothesis was that the negative mood state, induced by sad music, increases artistic creativity as compared with both positive and neutral mood states.

## 2. Materials and Methods

### 2.1. Participants

Ninety participants (55 women; mean age = 22.08 ± 2.33 years; age range = 18–30 years; mean education = 13.16 ± 1.94 years; education range = 8–17 years) were enrolled for this study from the Department of Biotechnological and Applied Clinical Sciences, University of L’Aquila. In order to calculate the minimum sample size, a power analysis was performed using G*Power 3.1.9.2 [36]. The analysis reflected the mixed factorial ANCOVA defined for the mood manipulation check. Thus, the following parameters were used: family test = F test analysis; statistical test = ANCOVA: fixed effects, main effects and interactions; the effect size was computed using the option ‘determine’, entering the value of 0.24 (see He et al. [37]—interaction effect ‘group x time’ with respect to valence), which returned an effect size of 0.05619515; alpha = 0.0001; power = 0.80 (minimum accepted, see Cohen [38]); numerator df = 2 [(3-1)*(2-1)]; the number of groups = 6 [(3 groups: positive, negative and neutral mood) × 2 (number of measurements: pre- and post-mood induction)]; the number of covariates = 1 (gender). The analysis suggested a minimum sample size of 90 participants, 30 participants per group. In the present study, 30 participants per group were used.

Thirty participants were randomly assigned to each mood condition (group):-Positive mood group (PM), 16 females, mean age = 22.07 ± 1.95;-Negative mood group (NM), 22 females, mean age = 21.90 ± 2.26;-Neutral mood (NeuM), 17 females, mean age = 22.27 ± 2.78.

Participants in the happy, sad and neutral conditions then selected music that was consistent with the condition to which they were assigned and listened to that music for 10 min. After filling out a short anamnesis questionnaire, all participants reported no neurological (e.g., trauma, brain injury, etc.) or psychiatric disorders (e.g., depression, anxiety, bipolar syndrome, eating disorders, as well as alcohol or drug addictions). Finally, no participant reported a background in visual art or creative activities. They signed the written informed consent. The study was designed in accordance with the ethical principles of human experimentation stated in the Declaration of Helsinki.

### 2.2. Materials

Musical excerpts. These latter consisted of musical tracks that each participant self-selected according to the pre-assigned condition (sad, happy, or neutral mood state). Ten titles of pieces of music per condition used by participants are listed in the Appendix A.

Clark’s Drawing Ability Test (CDAT) [39]. The CDAT was aimed at measuring the ability to create artistic drawings. Participants were asked to draw a front view of a house and a free drawing, instead of four drawings, as in the original CDAT (a house, a running person, a playground and free) in order to avoid fatigue. A pencil, coloured pencils and an eraser were also available. This short version of the CDAT was also used in previous studies [40].

The Profile of Mood States (POMS) [41,42]. The P.O.M.S. was aimed at assessing depression-dejection (D) and vigor-activity (V) mood states of participants before and after mood induction. It consists of 65 items, but 7 items were originally designed to measure ‘Friendliness’ (F), a factor that was proven psychometrically unsound. Thus, the typical POMS consists of 58 adjectives referring to mood states, including not only D and V but also tension-anxiety (T), anger-hostility (A), fatigue-inertia (S) and confusion-bewilderment (C). Each item was evaluated using a 5-point Likert scale, ranging from 0 ‘Not at all’ to 4 ‘extremely’. Although the predictions of the present study were based specifically on D and V, the other subscales, including the 7 adjectives related to F, were also administered to avoid tampering with reliability and validity [43,44].

### 2.3. Procedure

First, participants signed the written informed consent and filled in the anamnesis questionnaire. Afterwards, participants were randomly assigned to one of the following groups: positive, negative, or neutral mood state. Then, the POMS was administered for the first time. After completing the mood questionnaire, participants were asked to select and listen to their own music for 10 min. Participants were asked to select music according to the mood condition assigned, that is, happy, sad, or neutral, respectively. There were no specific instructions about the genre of the music: participants were free to select any kind of music if it evoked the pre-assigned mood state. Given that the majority of pieces of music last less than 10 min, participants were allowed to listen continuously to more than one piece of music. After 10 min, participants were instructed to fill in the POMS for the second time. Finally, they were instructed to draw both a front view of a house and a free drawing as creatively as possible. Fifteen minutes per drawing were given. The entire procedure lasted about 1 h.

### 2.4. Scoring of CDAT

Performance in the CDAT was evaluated using the Consensual Assessment Technique [45], which encompasses the idea that creativity products can be measured by an independent panel of judges, ideally but not necessarily experts in the field. Therefore, three independent and anonymous judges (mean age = 25.33; ±2.51; 2 females) blind to the experimental conditions evaluated drawings in terms of: (1) creativity, defined as original and appropriate ideas (that is involving novelty and relevance within the art context); (2) emotionality, defined as the extent to which the drawing elicited emotional responses, (e.g., smiling, calmness, tension and excitement; (3) aesthetic value, defined as beauty, pleasure and satisfaction on viewing. The judges were three psychology students attending a training (20 h) on theoretical models of creativity. They were shown examples of creative art for practice. Afterwards, judges began the evaluation session, separately rating each drawing provided by participants. For each parameter, the judges were asked to provide a single score using a 5-point Likert scale, ranging from 0 (not at all) to 4 (completely). The inter-rater correlation (intra-class correlation coefficient—absolute agreement) was significant for creativity (alpha = 0.92, *p* < 0.001), emotionality (alpha = 0.93, *p* < 0.001) and aesthetic value (alpha = 0.80, *p* < 0.005). For each parameter, the three ratings provided by the judges were averaged, leading to three scores for each drawing. Then, the final score for each parameter was obtained by averaging the scores of the two drawings.

### 2.5. Statistical Analysis

Statistical analyses were performed using SPSS Statistics version 20 for Windows (IBM Corporation, Armonk, NY, USA). Given that gender (0 = male; 1 = female) can affect mood states and creativity, a series of analyses of covariance were carried out after controlling for gender, as follows. First, two univariate ANCOVAs were performed in order to check differences between the NeuM, PM and NM groups in the V and D mood states prior to the mood induction. Second, two 3 × 2 mixed ANCOVAs were performed in order to check differences in V and D, using a between factor (group: NeuM, PM and NM) and a within factor (mood: pre- and post-induction). Third, three univariate ANCOVAs were performed in order to check group differences in creativity, emotionality and aesthetic value. For each significant ANCOVA, Bonferroni’s posthoc analysis was performed.

## 3. Results

The descriptive statistics are shown in Table 1 (mean and standard deviation for each variable).

### 3.1. Pre-Mood Manipulation Check

No significant difference in V and D mood states prior to the mood induction by group (happy, sad and neutral) [V: F(2,86) = 2555, *p* = 0.084; D: (2,86) = 3073, *p* = 0.051] was found. The covariate gender was not significant either in V [F(1,86) = 2051, *p* = 0.13] or in D [F(1,86) = 1771, *p* = 0.19]. This means that even though the three groups were statistically comparable in terms of the mood states under investigation before being involved in the mood induction procedure, the near-significant results require caution.

### 3.2. Mood Manipulation Check

The mixed ANCOVA performed on group (PM, NM, NeuM) x mood V (pre and post) showed a main effect of group [F(2,86) = 13.652; *p* < 0.001; partial ηp^2^ = 0.241] and a significant interaction group x mood V (F(2,86) = 18.230; *p* < 0.001; partial ηp^2^ = 0.300). The posthoc analysis (Bonferroni) showed that after the mood induction, the NM group reported lower scores than the PM and NeuM groups on the ‘vigour’ subscale, while the PM group felt more vigor than the NM group (*p*< 0.01). In addition, the NeuM and PM groups showed higher scores after than before mood induction, whereas the NeuM group showed lower scores after than before mood induction. The main effect of mood was not significant [F(1,86) = 0.380; *p* = 0.54] and the covariate gender [F(1,86) = 3.009; *p* = 0.090] (see Figure 1).

The mixed ANCOVA performed on group (PM, NM, NeuM) x mood D (pre and post) showed a main effect of group [F(2,86) = 16.831; *p* < 0.001; partial ηp^2^ = 0.281] as well as a significant interaction group x mood D [F(2,86) = 19.021; *p* < 0.001; partial ηp^2^ = 0.307]. The posthoc analysis (Bonferroni) showed that after the mood induction, the NM group reported higher scores than the PM and NeuM groups on the ‘depression’ subscale (*p* < 0.01). No difference was found between the PM and NeuM groups. In addition, the NM group showed higher scores after than before mood induction, whereas the NeuM group showed lower scores after than before mood induction. The main effect of mood D [F(1,86) = 0.562; *p* = 0.46] and the covariate gender [F(1,86) = 2.026; *p* = 0.16] were not significant (see Figure 2).

### 3.3. Mood Effects on Creativity, Emotionality and Aesthetic Value

The ANCOVAs showed a main effect of group, as follows: creativity F(2,86) = 6.519; *p* = 0.01; partial ηp^2^ = 0.132; emotionality F(2,86) = 6.904; *p* = 0.01; partial ηp^2^ = 0.138; aesthetic value F(2,86) = 2.848; *p* = 0.06. The post hoc analysis (Bonferroni) revealed that the NM group significantly yielded higher scores (*p* < 0.05) than the other two groups in creativity and emotionality but not in aesthetic value. No difference was found between the PM and NeuM groups (see Figure 3). The covariate gender was not significant in creativity [F(1,86) = 0.654; *p* = 0.42] and emotionality [F(1,86) = 0.101; *p* = 0.75], whereas it was significant in aesthetic value [F(1,86) = 4.067; *p* < 0.05; ηp^2^ = 0.045].

## 4. Discussion

In the current study, the role of positive and negative mood states induced by self-selected music was explored in visual artistic creativity in non-artists. Participants who listened to sad music produced more creative and emotional drawings than participants who listened to happy or neutral music. Given that the NM group reported higher scores than the PM and NeuM groups on the ‘depression’ subscale after mood induction, the negative mood state induced by music affected the relationships between music listening and artistic drawing. Although this result is not comparable with the effects of affective disorders [26], general neuroticism [46] and negative emotions [27] on artistic creativity, it shows that even a transient change in mood after listening to sad music can prime specific aspects of artistic creativity. This result does not depend on the individuals’ stereotype of the ‘tortured’ artist, which might have led participants who listened to sad music to consciously generate more creative and emotional drawings because they guessed this association. Indeed, participants were instructed and encouraged in all the three conditions to produce drawings that were as creative as possible.

Focusing on the theories reviewed above, this finding does not support the dual pathway model [6], specifically the idea that positive, activating mood states yield higher creativity because of enhanced flexibility as compared with deactivating mood states. In this study, participants primed to listen to sad music provided higher creativity and emotionality than participants primed to listen to happy music. This finding does not support the cognitive tuning model [9], that is, the idea that negative mood states lead to performing worse in a creativity task using a detailed-oriented information processing style. In this study, participants in a negative mood performed better in the artistic drawings. Following the mood as-input model [16], the present result is consistent with the idea that participants were informed that the task was not yet completed, and consequently persisted in working until the optimum level of creativity was achieved. Notably, participants had fifteen minutes per drawing, which allowed them to work until a satisfaction level was reached. Considering the hedonic contingency theory [10], the mood repair interpretation is also likely, if participants found the task interesting, and those who listened to negative music engaged more creatively in order to return to a neutral position.

However, another alternative interpretation might be based on the view that emotional ambivalence cued by self-selected music boosts artistic creativity. Simultaneous experience of mood states is perceived as an unusual affective state, signalling an atypical environment [47]. Consequently, an individual’s sensitivity to unusual associations can increase. Given that in the present experiment participants were instructed to self-select sad music, the possibility that while listening to music, they evoked not only sadness but also pleasure and enjoyment is likely. Interestingly, sadness related to listening to music is characterised by imagination, emotion regulation and empathy, with no real-life implications, and specifically, empathy plays a key role in modulating emotional responses to sad music, which can also convey an experience of pleasure [48]. Thus, listening to sad music also leads to experiencing nostalgia and peacefulness, whereas aesthetic appreciation and empathetic engagement are involved in enjoying sad music [49]. Importantly, sadness evoked by music can be pleasurable if perceived as non-threatening, aesthetically pleasing and producing psychological benefits based on autobiographical memories [50]. Therefore, it might be that in the present study, participants who self-selected sad music experienced emotional ambivalence, that is, a mixed mood state, that was more suitable for creativity and the emotionality of artistic creativity, providing more inputs for being creative as compared with the single positive mood evoked by listening to happy music. Notably, no significant effect of negative mood was found on the aesthetic value of artistic drawings. Probably, this result was due to the type of the instruction: participants were told to make creative rather than aesthetically pleasing drawings. This instruction plausibly primed the NM group towards creativity and emotionality.

The view that while listening to sad music participants experience emotional ambivalence, which increases subsequent creativity and emotionality, is plausible and intriguing, but needs to be confirmed. Future studies could better explore the effect of emotional ambivalence on creativity by evaluating not only the mood state change along different positive and negative dimensions but also the autobiographical memories associated with the listening, especially if letting participants use their favourite mood music. In addition, the key role of other variables related to individual differences might also be explored, such as personality and cognitive styles. The investigation of the mood ambivalence–creativity relationships can also be extended to other forms of creativity (e.g., verbal, musical, motor), considering a variety of parameters that define creativity (e.g., surprise, uniqueness, remoteness, cleverness, quality).

Some limitations need to be clarified. First, the music selected by participants was not evaluated in terms of valence and arousal. Although the mood manipulation check showed the effectiveness of the approach, future studies should better acknowledge the self-selected music in terms of valence and arousal dimensions. This would help to clarify the extent to which the valence and arousal of the music samples influenced the moods of participants, and, consequently, the different mediation roles of these two dimensions in the relationship between music listening and artistic drawing. Indeed, given that participants self-selected music based on their personal preference, it cannot be assumed that a sad mood (defined by negative valance and lowarousal) is related to negative valenced and low arousing music, or a happy mood (defined by positive valence and high arousal) to positive valenced and high arousing music. In other words, it might be that individuals got into a sad mood by listening to negative and high-arousing music, or a happy mood by listening to positive and low-arousing music. Additionally, the extent to which music with lyrics evoked the pre-assigned mood as compared with instrumental music was not checked out. Previous studies showed that music with lyrics strongly evokes sadness or negative emotions, whereas instrumental music plays a stronger role in evoking happiness (e.g., [51]). Probably this effect occurs because music with lyrics is more personalised and is better associated with memories and sadness, whereas instrumental music has a more immediate effect on emotions with its acoustical parameters and might quickly affect self-reports [52]. Therefore, although the happy group reported higher scores on the ‘vigour’ scale than the sad group, which in turn reported higher scores on the ‘depression’ scale than the other two groups, future research should better explore the issue of melody- and lyrics-based music in evoking positive and negative moods. Furthermore, even though no significant difference in V and D mood states prior to mood induction was found, the analysis revealed near-significant results after controlling for gender. This means that the role of gender cannot be neglected in future research. Finally, in the current study, the time of day subjects participated in the experiment was not recorded. Notably, previous studies [53] reported circadian variations in mood states as well as performance differences due to the time of the day and the individual’s circadian typology. Further studies should consider these factors to explore the role of mood in creative production.

To conclude, negative mood states have the potential to affect creativity positively. The theoretical models incorporating negative mood states are rare and need further development to explain the negative mood–creativity relationships.

## Figures and Tables

**Figure 1 brainsci-13-00149-f001:**
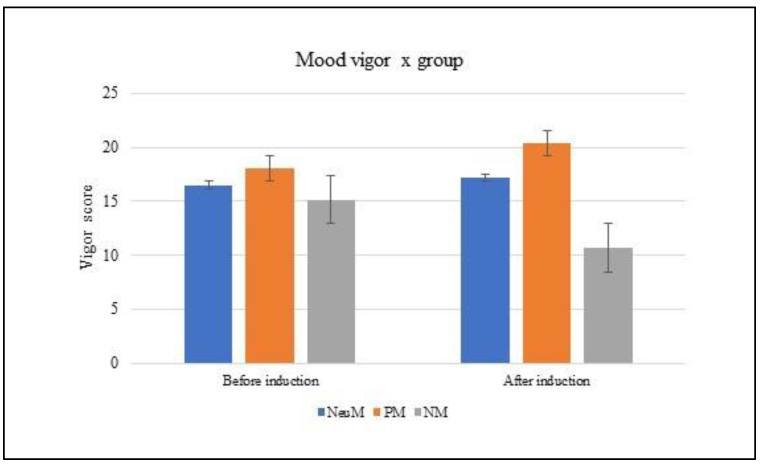
Rating of the ‘vigor’ sub-score of the POMS before and after mood induction, with respect to positive, negative and neutral music groups. The error bars represent the standard errors of the means.

**Figure 2 brainsci-13-00149-f002:**
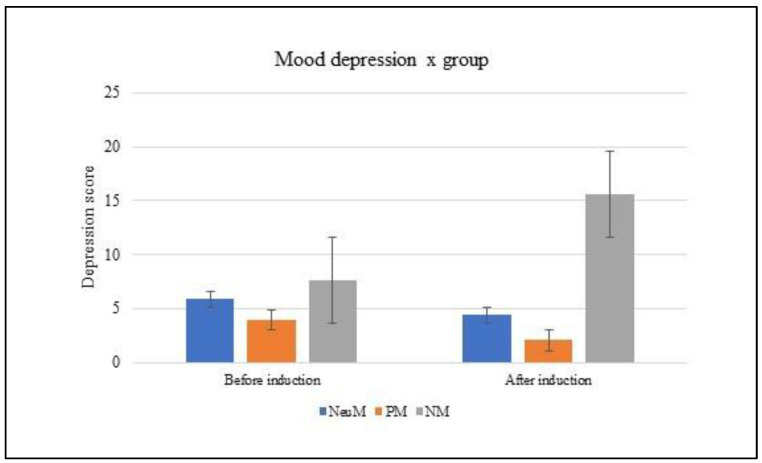
Rating of the ‘depression’ sub-score of the POMS before and after mood induction, with respect to positive, negative and neutral music groups. The error bars represent the standard errors of the means.

**Figure 3 brainsci-13-00149-f003:**
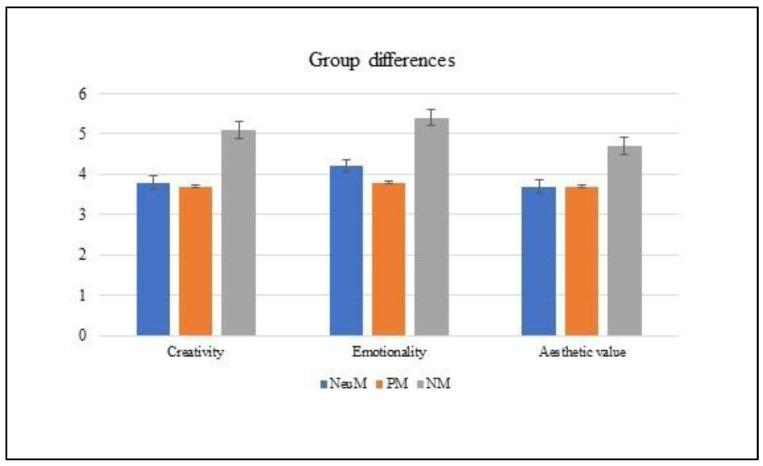
Rating of the creativity, emotionality and aesthetic value of the artistic drawings with respect to positive, negative and neutral music groups. The error bars represent the standard errors of the means.

**Table 1 brainsci-13-00149-t001:** Descriptive statistics.

	PM Group	NM Group	NeuM Group
**Vigor before induction**	18.03 (5.82)	15.17 (6.13)	16.50 (4.22)
**Vigor after induction**	20.40 (5.54)	10.67 (6.58)	17.20 (4.14)
**Depression before induction**	4.00 (4.33)	7.63 (7.37)	5.90 (6.44)
**Depression after induction**	2.10 (3.36)	15.60 (11.58)	4.43 (6.27)
**Creativity**	3.70 (1.39)	5.10 (1.91)	3.79 (1.41)
**Emotionality**	3.84 (1.43)	5.37 (1.88)	4.22 (1.43)
**Aesthetic value**	3.66 (1.73)	4.69 (2.00)	3.66 (1.25)

## Data Availability

The dataset will be made available upon request.

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
