# Peer review of "The Effect of Sadness on Visual Artistic Creativity in Non-Artists"

_brainsci, 2023, doi:10.3390/brainsci13010149_

Round 1
Reviewer 1 Report
I really enjoyed reading the manuscript, which provides a very thorough and interesting insight into the role of mood in creative performance.
My only small remark/question is related to the music selected by participants. Namely, I was wondering whether the Authors evaluated the music on dimensions such as valence and arousal - much as I understand that participants selected it based on their own (subjective) opinions, I can imagine a huge variability in the music chosen, which might have confounded the results obtained. Also, the presence/absence of lyrics might have been an additional confound - i.e. if participants in the sad mood condition chose a higher proportion of instrumental music (with no lyrics), they might have been able to focus more on the task itself due to a limited access to phonological and lexico-semantic distractors (which might have, on the other hand, influenced participants in the neutral/positive mood conditions).
Author Response
I really enjoyed reading the manuscript, which provides a very thorough and interesting insight into the role of mood in creative performance.
My only small remark/question is related to the music selected by participants. Namely, I was wondering whether the Authors evaluated the music on dimensions such as valence and arousal - much as I understand that participants selected it based on their own (subjective) opinions, I can imagine a huge variability in the music chosen, which might have confounded the results obtained. Also, the presence/absence of lyrics might have been an additional confound - i.e. if participants in the sad mood condition chose a higher proportion of instrumental music (with no lyrics), they might have been able to focus more on the task itself due to a limited access to phonological and lexico-semantic distractors (which might have, on the other hand, influenced participants in the neutral/positive mood conditions).
Response: We acknowledged these suggestions in the last part of the discussion as limits of the study, as follows: ‘Some limitations need to be clarified. First, music selected by participants was not evaluated in terms of valence and arousal. Although the mood manipulation check showed the effectiveness of the approach used, future studies should better acknowledge the self-selected music in terms of valence and arousal dimensions. Additionally, the extent to which music with lyrics evoked the pre-assigned mood as compared to instrumental music was not checked out. Previous studies showed that music with lyrics strongly evokes sadness or negative emotions, whereas instrumental music plays a stronger role in evoking happiness [e.g., 50]. Probably this effect occurs because music with lyrics is more personalized and is better associated with memories and sadness, whereas instrumental music has a more immediate effect on emotions with its acoustical parameters and might quickly affect self-reports [51]. Therefore, although the happy group reported higher scores on the ‘vigour’ scale than the sad group, which in turn reported higher scores on the ‘depression’ scale than the other two groups, future research should better explore the issue of melody- and lyrics-based music in evoking positive and negative moods’.
Reviewer 2 Report
The manuscript provides an interesting study on the effect of sadness on visual artistic creativity in non-artistic participants. The sample of participants is adequate, made up of both sexes, although with a higher proportion of women. Below I go on to detail the aspects that the authors should review to improve the present version of the manuscript:
- The introduction is excessively long for an experimental study, especially the last two paragraphs on page 3 that specify the aim should be reduced without This supposes a loss of the necessary information.
– The analyzes should be carried out again including the gender factor, there are enough participants in each group of men and women for this. This approach is necessary because the influence of sex on mood disorders and modo states is well known, as well as on the performance of all kinds of tasks.
– It is not clear when it is specified that all participants were healthy and had no addiction what is considered healthy (since addiction is a disorder). Have medical and psychiatric pathologies been ruled out? This information must be provided in a somewhat more precise way.
– It is not mentioned whether the time of day of the recordings was controlled and this is of vital importance because there are circadian variations both in mood states and in performance and differences between people due to the individual difference in circadian typology (more marked in young people). In this sense, if the time was controlled, it must be specified and, in the event that it was not controlled, indicate it in the discussion section as a limitation to be overcome in the future. In both cases, it is suggested to incorporate the pioneering papers in this regard: https://pubmed.ncbi.nlm.nih.gov/8131672/, https://pubmed.ncbi.nlm.nih.gov/8131672/ There are other limitations of the study, which the authors should mention in association with any proposal to overcome them in the future.
– It is advisable to create a section in the Materials and methods of statistical analysis, which would facilitate the presentation and structuring of the work.
– Figure 2 cannot be presented with continuous lines since the three groups are made up of different participants. It is appropriate to use a histogram or a box-plot.
Author Response
- The introduction is excessively long for an experimental study, especially the last two paragraphs on page 3 that specify the aim should be reduced without This supposes a loss of the necessary information.
Response: We thanks the reviewer 2 for this suggestion. The introduction has been streamlined, excluding unnecessary information and repetitions, and now it is more readable.
– The analyzes should be carried out again including the gender factor, there are enough participants in each group of men and women for this. This approach is necessary because the influence of sex on mood disorders and mood states is well known, as well as on the performance of all kinds of tasks.
Response: The analyses were performed again after controlling for gender. We highlighted the changes. Overall the results did not change, therefore the discussion was still supported by the analyses. Please consider that we updated the power analysis using the Ancova as a model for computing the power. We also included the descriptive analysis.
– It is not clear when it is specified that all participants were healthy and had no addiction what is considered healthy (since addiction is a disorder). Have medical and psychiatric pathologies been ruled out? This information must be provided in a somewhat more precise way.
Response: In order to avoid confusion with the meaning of the term ‘healthy’, we dropped out the term ‘healthy’ and specified that participants reported no neurological (e.g., trauma, brain injury, etc...) or psychiatric disorder (e.g., depression, anxiety, bipolar syndrome, eating disorders, etc...), and no alcohol or drug addiction by a short anamnesis questionnaire.
– It is not mentioned whether the time of day of the recordings was controlled and this is of vital importance because there are circadian variations both in mood states and in performance and differences between people due to the individual difference in circadian typology (more marked in young people). In this sense, if the time was controlled, it must be specified and, in the event that it was not controlled, indicate it in the discussion section as a limitation to be overcome in the future. In both cases, it is suggested to incorporate the pioneering papers in this regard: https://pubmed.ncbi.nlm.nih.gov/8131672/, https://pubmed.ncbi.nlm.nih.gov/8131672/
Response: We did not consider this issue, that is the time the day subjects participated in the experiment was not recorded. However, we highlighted it as a limit in the discussion.
- There are other limitations of the study, which the authors should mention in association with any proposal to overcome them in the future.
Response: We added a paragraph highlighting the limits of the study, as follows: ‘Some limitations need to be clarified. First, music selected by participants was not evaluated in terms of valence and arousal. Although the mood manipulation check showed the effectiveness of the approach used, future studies should better acknowledge the self-selected music in terms of valence and arousal dimensions. Additionally, the extent to which music with lyrics evoked the preassigned mood as compared to instrumental music was not checked out. Previous studies showed that music with lyrics strongly evokes sadness or negative emotions, whereas instrumental music plays a stronger role in evoking happiness [e.g., 50]. Probably this effect occurs because music with lyrics is more personalized and is better associated with memories and sadness, whereas instrumental music has a more immediate effect on emotions with its acoustical parameters and might quickly affect self-reports [51]. Therefore, although the happy group reported higher scores on the ‘vigour’ scale than the sad group, which in turn reported higher scores on the ‘depression’ scale than the other two groups, future research should better explore the issue of melody- and lyrics-based music in evoking positive and negative moods. Furthermore, even though analysis on the covariate gender revealed near-significant results, no gender differences should be confirmed in future research. Finally, in the current research, the time of the day subjects participated in the experiment was not recorded. Notably, previous studies [52] reported circadian variations in mood states as well as performance differences due to the time of the day and the individual’s circadian typology. Further studies should consider these factors to explore the role of mood in creative production’.
– It is advisable to create a section in the Materials and methods of statistical analysis, which would facilitate the presentation and structuring of the work.
Response: The section has been added.
– Figure 2 cannot be presented with continuous lines since the three groups are made up of different participants. It is appropriate to use a histogram or a box-plot.
Response: The figures were changed, now they are represented as histograms.
Round 2
Reviewer 2 Report
The authors have made the changes according to my main concerns and suggestions, which has greatly improved this second version of the manuscript.
Author Response
The authors have made the changes according to my main concerns and suggestions, which has greatly improved this second version of the manuscript.
Response: Thank you very much fro your precious advices.